# Biomarkers of Migraine: An Integrated Evaluation of Preclinical and Clinical Findings

**DOI:** 10.3390/ijms24065334

**Published:** 2023-03-10

**Authors:** Chiara Demartini, Miriam Francavilla, Anna Maria Zanaboni, Sara Facchetti, Roberto De Icco, Daniele Martinelli, Marta Allena, Rosaria Greco, Cristina Tassorelli

**Affiliations:** 1Department of Brain and Behavioral Sciences, University of Pavia, Via Bassi 21, 27100 Pavia, Italy; 2Unit of Translational Neurovascular Research, IRCCS Mondino Foundation, Via Mondino 2, 27100 Pavia, Italy

**Keywords:** migraine, CGRP, endocannabinoid system, inflammation, biofluids

## Abstract

In recent years, numerous efforts have been made to identify reliable biomarkers useful in migraine diagnosis and progression or associated with the response to a specific treatment. The purpose of this review is to summarize the alleged diagnostic and therapeutic migraine biomarkers found in biofluids and to discuss their role in the pathogenesis of the disease. We included the most informative data from clinical or preclinical studies, with a particular emphasis on calcitonin gene-related peptide (CGRP), cytokines, endocannabinoids, and other biomolecules, the majority of which are related to the inflammatory aspects and mechanisms of migraine, as well as other actors that play a role in the disease. The potential issues affecting biomarker analysis are also discussed, such as how to deal with bias and confounding data. CGRP and other biological factors associated with the trigeminovascular system may offer intriguing and novel precision medicine opportunities, although the biological stability of the samples used, as well as the effects of the confounding role of age, gender, diet, and metabolic factors should be considered.

## 1. Introduction

### Migraine Pathogenesis

Migraine is a complex disease characterized by recurring attacks with unilateral or bilateral head pain, often pulsating in quality, of moderate to severe intensity, associated with other disabling symptoms, and aggravated by physical activity. According to the Global Burden of Diseases, migraine represents the third most frequent and the second most disabling condition in humans, affecting about 16% of the world’s population [1].

The latest edition of the International Classification of Headache Disorders (ICHD-3) identifies several subgroups of migraine based on the associated symptoms and the number of monthly headache days [2,3]. The majority of migraine patients experience migraine without aura and with a frequency < 15 days per month (henceforth called episodic migraine (EM)). About 3% of migraine patients experience more than 15 headache days/month for at least 3 months and qualify as chronic migraine (CM) [3]. It has been estimated that each year, 3% of EM subjects transition to CM, frequently as a result of ineffective acute treatment and/or acute drug overuse or because of bearing a particularly aggressive type of migraine [4]. When the transition from EM to CM is associated with the overuse of acute medications, patients should also receive the diagnosis of medication overuse headache (MOH) [2]. In all patients, migraine attacks occur with a cyclic pattern of different phases: a pain-free interictal phase, a prodromal phase with changes in perception and behavior, an ictal phase with headache and other associated symptoms, and a post-ictal phase without headache [5]. Additionally, before the ictal phase, about one-third of patients with migraine experience migraine aura, which is described by the ICHD-3 as brief, entirely reversible positive or negative neurological symptoms occurring alone or in succession [2]. Regarding the pathophysiology of a migraine attack, it is believed to originate from the activation of the brainstem and diencephalic nuclei, followed by the involvement of the trigeminovascular system (TVS) [6,7,8]. When the TVS is activated, vasoactive peptides, such as calcitonin gene-related peptide (CGRP), and proinflammatory mediators are released in the meninges, resulting in a condition of dural neurogenic inflammation and central sensitization [9].

Several clinical and preclinical models of migraine pain have been used to replicate and study the mechanisms underlying the pathophysiology of migraine pain, focusing on trigeminal sensory processing and the role of vascular and neuronal components involved in the disease. These clinical and/or preclinical models include (i) infusion or the systemic administration of migraine-inducing compounds such as nitroglycerin (NTG), CGRP, and pituitary adenylate cyclase-activating peptide-38 (PACAP-38) [10,11,12]; (ii) chemical (with inflammatory molecules or irritant substances) or electrical activation of the structures included in the TVS (e.g., meninges and trigeminal ganglia) [12,13,14]; and (iii) the induction of cortical spreading depression (CSD) over the cortex surface, which is considered as the neurophysiological correlate of migraine aura [15].

Besides the utility of unraveling the pathophysiological mechanisms of migraine, the use of these clinical and preclinical models may contribute to the identification of biomarkers with diagnostic or prognostic value [16,17]. Advances in the identification of migraine-specific biomarkers are indeed essential for improving the diagnosis and treatment of migraine, fostering the application of precision medicine strategies.

The research on migraine biomarkers has covered different fields, ranging from genetics, neuroimaging, and biochemical approaches. Systematic reviews and meta-analyses have failed so far to identify reliable candidate biomarkers in peripheral blood (plasma/serum, saliva) or in the cerebrospinal fluid (CSF) of patients with EM or CM [18,19,20,21]. Several reliable experimental models of migraine are available in humans and animals. These have been extensively used in the past two decades [10,12]. In this review, we summarized data from the most studied potential biomarkers for migraine, also including in the analysis the comparative output of human and animal models, wherever available. Our ultimate aim was to further elucidate the neurobiology linking the potential biomarkers to migraine pathophysiology.

## 2. Methods

In this narrative review, the data originating from both clinical and preclinical studies on migraine were retrieved from the PubMed/MEDLINE database, covering the period from September 1988 to November 2022.

The search methodology included studies conducted on adult patients (pediatric migraine was not considered) suffering from EM (either with or without aura) or CM (including the subgroup with MOH). Regarding preclinical studies, the articles included were those involving animal models (either with rats or mice) that reproduce one or more pathophysiological features of migraine pain. Of these clinical and preclinical studies, only those reporting changes in circulating (i.e., in blood, CSF, and urine) biomarkers were included.

## 3. The Potential Circulating Biomarkers Described in Migraine Patients and Migraine Animal Models

### 3.1. Neuropeptides

In this section, we discuss the main (neuro)peptides differently involved in the modulation of migraine-related structures and considered to be of clinical relevance, and glutamate, the principal excitatory neurotransmitter within the central nervous system [22]. CGRP, substance P (SP), PACAP, vasoactive intestinal peptide (VIP), and neuropeptide Y (NPY) are implicated in craniocervical vasodilatation, with SP also having a role in plasma protein extravasation and CGRP and PACAP in peripheral and/or central sensitization, the main mechanisms related to migraine pathophysiology [23], whereas glutamate has been linked to neuronal hyperexcitability and plays a pivotal role in triggering migraine attacks [24].

#### 3.1.1. Calcitonin Gene-Related Peptide (CGRP)

CGRP is a neuropeptide belonging to the calcitonin family; it exists in two isoforms, α and β, encoded by two different genes, CALCA and CALCB, respectively [25]. The α form is mainly expressed in the central and peripheral nervous system, whereas the β form is located predominantly in the enteric nervous system. Notwithstanding the major role of CGRP in the pathophysiology of migraine [26,27], its status as a biomarker for migraine diagnosis is still controversial. A huge number of studies have investigated CGRP levels in different migraine subtypes and multiple biological specimens (plasma, serum, saliva, and CSF), yielding contrasting data, which so far prevents considering CGRP a reliable migraine biomarker.

Multiple studies conducted in patients assessed during the interictal phase report higher plasma/serum/saliva/CSF CGRP levels in EM and CM patients than in healthy controls (HC) [28,29,30,31,32,33,34,35,36,37,38,39,40,41]. Differently from EM and HC, Cernuda-Morollón and colleagues found elevated CGRP serum levels in CM women with and without acute medication overuse, suggesting a potential pathophysiological mechanism for CGRP in migraine chronification [35]. This finding was confirmed by Pérez-Pereda and colleagues [42]. Interestingly, a positive correlation between serum CGRP levels and pain intensity was also reported [41]. It must, however, be noted that other studies did not report any changes in CGRP levels among EM and CM patients [42,43,44,45,46].

When considering the ictal phase, migraine patients show higher serum levels of CGRP than HC [47]. Some studies clearly show an increase in plasma/saliva CGRP levels during the attack compared with the interictal period in EM patients [34,40,48,49,50]. By contrast, other studies reported no difference in CGRP levels in jugular venous blood between the ictal and interictal periods in EM subjects [45,51]. Similarly, Cady and colleagues showed no difference in the CGRP saliva levels of CM patients during a migraine attack compared with EM patients, suggesting that the threshold for central activation from peripheral input is possibly lower in CM than in EM [52].

When considering the aura subtype of migraine, an increase in serum CGRP levels was found in CM with aura patients compared with those without aura [35], while no difference between EM with and without aura was reported in patients evaluated in the interictal phase [32,34,35] and the ictal phase [47]. Hence, further studies on the importance of CGRP for aura phenomena and migraine are needed to verify whether the peripheral levels of the neuropeptide may have an informative role in migraine diagnosis [53].

It is interesting to note that CGRP plasma levels also change in experimentally induced migraine. EM patients challenged with NTG develop migraine-like attacks associated with CGRP plasma levels increase compared with the baseline [54]; similarly, an increase in CGRP serum/plasma levels was reported in animals subjected to NTG challenge in a migraine animal model [55,56]. Changes in peripheral CGRP levels have also been found in animal migraine models based on the electrical stimulation of the superior sagittal sinus, trigeminal ganglion, or dura mater, in an animal model of intracisternal inflammatory soup and in mice with RAMP-1 deficiency [57,58,59,60,61,62]. By contrast, experimental human models (e.g., hypoxia and VIP infusion) did not report any significant change in CGRP blood plasma levels in migraine patients [46,63]. Finally, in both human and animal models, the increased CGRP levels were reduced by migraine treatments [55,56,60,64]. In agreement, CGRP levels were found to be significantly lower after treatment with onabotulinumtoxinA in CM patients [52,65] and after detoxification in those with MOH [66]. Furthermore, it seems that peripheral CGRP levels may be useful in predicting some treatments’ outcomes. Higher baseline CGRP levels in EM patients were associated with better response to rizatriptan [49] and erenumab [67]. Similarly, the baseline levels of CGRP in patients with CM and MOH who benefitted from the withdrawal of overused drugs were also significantly higher than those detected in non-responders to the withdrawal procedure [66]. Cernuda-Morollón and colleagues [36] reported a 28-fold higher probability to respond to onabotulinumtoxinA for CM with a serum CGRP level above the threshold of 72 pg/mL. Dominguez and colleagues demonstrated that a CGRP level >50 ng/mL in the peripheral blood is associated with a good response to onabotulinumtoxinA treatment in CM patients [68].

According to recent meta-analysis studies, circulating CGRP is linked to the pathophysiology of migraine, and its peripheral level may be used as a biomarker for migraine diagnosis and as a potential indicator of treatment efficacy [20,69], notwithstanding a certain level of heterogeneity. Indeed, the meta-analyses show higher concentrations of CGRP in the CSF and blood of CM or EM patients compared with HC, as well as in the ictal phase of EM compared with the interictal period. Age, aura, menstrual cycle, frequency of migraine attacks, medication overuse, participant selection, psychological factors, and differences in blood from the jugular vein or antecubital may all have been potential confounders contributing to the heterogeneity of results [20,70].

#### 3.1.2. Other Peptides of the CGRP Family

Calcitonin, adrenomedullin, and amylin, as well as their receptors, are present within the TVS but differ in expression and localization [71]. The levels of pro-calcitonin and other members of the calcitonin family of peptides were altered in subjects with migraine [72]. Specifically, procalcitonin serum levels were higher in EM patients during the ictal phase when compared to the interictal period and HC [73]. The infusion of adrenomedullin for 20 min resulted in migraine attacks in 55% of migraine patients, whereas placebo infusion resulted in migraine attacks in only 15% of the patients [74].

Like CGRP, amylin belongs to the calcitonin peptide family; it is produced and secreted by β cells in the pancreas and shares some CGRP receptors and biological activities [75]. This neuropeptide and its receptors are found in migraine pain-related structures [76]. In a single study, amylin plasma levels were higher during the ictal phase in CM patients than in EM patients and HC [77], thus suggesting a possible role in disease chronicization [75]. However, additional studies are needed to test the prognostic/diagnostic potential of the neuropeptide. Preclinical experiments investigating the symptoms associated with migraine showed that amylin treatment causes cutaneous hypersensitivity and light aversion in mice [74]. In agreement, pramlintide infusion, an amylin analog, caused migraine-like symptoms in migraine sufferers without aura, probably via its potent activity on the CTR/RAMP complexes. Amylin and pramlintide are weak agonists of the CGRP receptor, with potencies 100 times lower than CGRP at the conventional CGRP receptor [78].

CGRP can bind to both the canonical CGRP receptor and the non-canonical CGRP receptor (AMY1); thus, an intriguing question is whether the CGRP receptor alone is the essential molecular site for anti-migraine therapy, or whether the AMY1 receptor is also involved in migraine pathophysiology, with amylin and/or CGRP being the primary ligand agonists [78].

#### 3.1.3. Substance P (SP) 

The neuropeptide SP belongs to the tachykinin family. It is abundant in the TVS, where it probably contributes to the transmission of pain. SP is a vasodilator, and its primary role in migraine is associated with plasma protein extravasation and vasodilation during TVS activation [6].

Plasma SP levels were higher in EM patients evaluated during the interictal phase when compared with HC [32], without differences between EM with and without aura [32]. CM patients showed higher plasma and salivary SP levels than HC patients, with a correlation between SP levels and pain intensity [33]. Nicolodi and Del Bianco [79] reported an increase in salivary SP levels during migraine attacks compared with HC, but other authors failed to detect differences in SP plasma in EM subjects compared with HC [80,81].

Interestingly, the stimulation of the trigeminal ganglion in humans induced an increase in SP plasma levels [82], a finding that has more recently been confirmed in the animal model based on dural electrical stimulation in rats [62]. Thus, it seems that, at least when detected in the plasma, SP may be a candidate biomarker of migraine. It must, however, be noted that SP antagonists failed to prove efficacy in migraine treatment [83].

#### 3.1.4. Pituitary Adenylate Cyclase-Activating Polypeptide (PACAP)

PACAP is a member of the glucagon/secretin superfamily that can be found in two forms (PACAP-27 and PACAP-38). The isoform PACAP-38 predominates in neuronal tissues; it is found in parasympathetic and sensory neurons, such as migraine-relevant brain structures, where it modulates pain processing and has vasodilatory effects [84].

PACAP-38 serum levels were higher in CM subjects tested interictally than in EM and HC patients, without any difference between EM and HC [42]. By contrast, Cernuda-Morollón et al. [85] failed to detect any differences in the serum PACAP-38 levels of subjects with CM or EM tested interictally and compared with HC. In this context, it is worth noting that Togha et al. [86] found higher interictal serum levels in EM patients than in CM and HC patients. In this variable scenario, some authors also reported lower interictal PACAP-38 plasma levels in EM than in HC [87,88], although PACAP-38 levels tended to increase in the ictal phase [87,89] without, however, reaching significantly different levels than HC [87]. Zagami et al. [89] found higher PACAP levels in the external jugular vein of patients with moderate or severe pain intensity; the levels decreased 1 h after receiving sumatriptan medication and further upon the end of the episode. These results are consistent with preclinical models in which PACAP-38 plasma levels increased after the electrical stimulation of the superior sagittal sinus, trigeminal ganglion, or dura mater [57,61,62,90].

Multiple studies have confirmed the function of PACAP in the pathophysiology of migraine [84]. PACAP infusion has been shown to induce a migraine-like attack in migraine patients [91]. The intravenous administration of PACAP may cause CGRP release and migraine attacks [89]. However, the contradictory data presented above do not speak in favor of the suitability of PACAP as a biomarker of migraine diagnosis or progression. As highlighted by other authors, sensitive plasma tests and improved collection techniques are required to evaluate the relevance of this neuropeptide in migraine pathogenesis [92].

#### 3.1.5. Vasoactive Intestinal Polypeptide (VIP)

VIP, like PACAP, is a polypeptide belonging to a glucagon/secretin superfamily. It is expressed in the parasympathetic nerves and exerts vasodilatory effects on cerebral and cortical pial vessels [93].

Increased interictal VIP levels were found in CM compared with EM and HC patients [36,42,65,85], and between EM and HC [31,65,86], but not in all studies [42]. An increase in VIP plasma levels was reported in animal models of migraine based on the electrical stimulation of the trigeminal ganglion or dura mater [61,62]. By contrast, no difference was found in plasma VIP levels collected from the cubital fossa and external jugular vein, during the headache phase in EM patients without and with aura compared with control values. However, two patients with aura showed prominent symptoms of lacrimation and rhinorrhea with marked elevations in the external jugular vein VIP levels [81].

Interestingly, changes in VIP levels were found in migraine patients with pronounced autonomic symptoms [81,94]; indeed, it was found that VIP correlates with the presence and degree of cranial parasympathetic symptoms in CM [95], thus suggesting a link between VIP levels and the degree of the activation of the cranial parasympathetic system in migraine [96]. VIP plasma levels were significantly decreased after rizatriptan, suggesting its potential use as a therapeutic biomarker [94]. In patients with EM, VIP causes migraine attacks 2 h after its infusion, suggesting an important role in migraine pathophysiology [97]. Increased plasma levels of CGRP were found in EM before the onset of migraine attacks but were unrelated to the occurrence of VIP-triggered migraine attacks [46]. The continuous intravenous infusion of VIP over two hours causes delayed mild headaches in HC, as well as long-lasting cranial vasodilation and activation of the cranial parasympathetic system [95,96]. VIP receptors are localized in rat middle meningeal artery [98], and the antagonism of the VPAC1 receptor represents a potential target for migraine headaches. VPAC1 and VPAC2 play a role in the stimulation of parasympathetic cerebral outflow during migraine attacks [99].

These observations sparked interest in elucidating the function of VIP. Future research is required to unveil the mechanisms underlying VIP-induced migraine attacks and the potential utility of VIP as a possible biomarker.

#### 3.1.6. Neuropeptide Y (NPY)

NPY is found in the sympathetic nerve endings innervating the dura mater and pial blood vessels and cerebral arteries, and it acts as a vasoconstrictor [23].

Higher CSF NPY levels have been found in subjects with migraine during the ictal period compared with HC [100]. In accord, elevated NPY plasma levels were reported in an animal model of migraine after the electrical stimulation of the trigeminal ganglion [61]. On the other hand, some investigations did not detect any change in NPY plasma or CSF levels either during or outside of a migraine attack in migraine patients [81,101] or even reported lower levels of NPY in subjects with migraine than in HC [102].

### 3.2. Classic Neurotransmitters

Some authors have proposed that migraine is linked to the altered metabolism of glutamate, serotonin, gamma-aminobutyric acid, dopamine, and noradrenalin [103,104]. In support, alterations in their precursors and metabolites, as well as in the neurotransmitters themselves, have been reported in some clinical and preclinical studies [103,104,105,106,107,108,109,110]. A recent meta-analysis showed that patients with EM have higher 5-HT blood levels than HC patients, both ictally and interictally, although the authors reported a substantial heterogeneity across studies [20]. However, serotonin plasma levels did not change after NTG in rats [111].

Clinical and preclinical observations suggest an involvement of the excitatory neurotransmitter glutamate in migraine mechanisms. For instance, glutamate is implicated in TVS activation, as in central sensitization [24,112,113]. In migraine, the glutamatergic system becomes overactive. According to different studies [106,114,115,116,117,118,119], EM and CM patients had considerably greater plasma and salivary levels of glutamate than HC during the headache-free time. In addition, the baseline glutamate levels in the CM group decreased following preventive treatment (regardless of the type of preventive drug) but were still higher than HC [117]. Notably, Nam et al. [118] described higher salivary glutamate levels in CM than in EM.

The levels of glutamate further increased in patients with migraine in the ictal phase [114]; within the ictal phase, higher levels of glutamate in CSF were found in EM and CM than in HC [30,120,121]. In agreement with clinical findings, increased plasma glutamate levels were observed in an animal model of migraine based on NTG administration [122].

The above-mentioned data are supported by a meta-analysis [69] pointing to circulating glutamate levels as a putative biomarker to discriminate between migraine patients and HC and possibly also to differentiate CM from EM.

### 3.3. Mediators of Inflammation and Immunity

Migraine is associated with an alteration in peripheral immune homeostasis, inflammation, and autoimmune diseases. According to some research on migraine patients, inflammatory mediators may decrease the threshold for the onset of the attack, leading to central sensitization and promoting the persistence and progression of migraine [123,124]. As a result, a widespread alteration in inflammatory patterns is observed in migraine sufferers, supporting the hypothesis that inflammation plays a role in the evolution of the disease [125,126,127,128,129].

Here, we will focus on the most studied inflammatory mediators that may qualify as potential migraine biomarkers.

#### 3.3.1. Cytokines

Although there may be some differences, it is clear that changes in inflammatory cytokines play a role in migraine. The findings related to cytokine changes in migraine have been the object of recent overviews [130,131]. Similarly, the C-reactive protein (CRP), a positive acute-phase protein that increases in response to inflammation, might play a role in migraine pathogenesis, but the findings are conflicting [132,133,134,135]. In this section, we focus on the serum/plasma or CSF levels of tumor necrosis factor-alpha (TNF-α), interleukin-1 beta (IL-1β), and interleukin-6 (IL-6) because of their crucial role in trigeminal pain [130,136,137].

Specifically, the levels of proinflammatory cytokines such as IL-6, TNF-α, IL-1β, and transforming growth factor-β1 (TGF-β1) were elevated in EM during the interictal phase [132,138,139,140]. Interestingly, IL-6, TNF-α, and IL-1β levels were also increased in the serum of rats treated with NTG [56]. The studies focusing on the ictal phase of migraine report a wide variability in cytokine levels, which were either stable or changing compared with interictal values, indicating the dynamic nature of these inflammatory molecules [141,142]. Such contrasting results may be related to preanalytical sampling procedures. For instance, anticoagulants or specific clock proteins in the circadian system may interfere with assays [143]. Some studies suggest that the serum levels of IL-6 and TNF-α are higher in CM patients than in EM patients [130,134,144], although Rozen and Swidan described a substantial rise in TNF-α levels in the CSF, but not serum, of CM patients when compared to HC [145].

To summarize, the relationship between changes in TNF-α and other cytokines and migraine pathogenesis is uncertain. The findings of numerous studies [135] are diverse and occasionally contradictory, which limits the possibility of including these cytokines in the panel of biomarkers for disease diagnosis.

#### 3.3.2. Adipocytokines

Obesity is listed among the risk factors for migraine and its chronification [146]. As a result, the involvement of adipocytokines in migraine would not be surprising. Several studies suggest that CM patients have higher levels of serum leptin and adiponectin than EM patients and HC [147,148,149]. Interestingly, the ictal serum levels of adipokines in EM were associated with pain severity and treatment response [150]. Although still little exploited, this line of research seems to hold promise for the possible identification of biomarkers for the risk of the negative outcomes of migraine disease.

#### 3.3.3. Prostaglandins

Mounting evidence from clinical and preclinical data supports the involvement of prostaglandins in migraine pathophysiology [151]. Prostaglandins and their receptors are widely distributed within the trigeminovascular structures, thus highlighting their role in the trigeminal pain pathways [151]. Estrogen fluctuations cause changes in prostaglandin production. An increased release of prostaglandins during the perimenstrual period leads to perimenstrual pain and increased proneness to migraine [152,153]. Conversely, estrogen withdrawal may increase vulnerability to prostaglandins and stimulate neuroinflammation via the increased production of neuropeptides such as CGRP, SP, and neurokinin [154]. Prostaglandin infusion causes headaches and the dilatation of intra-cranial and extra-cranial arteries in migraine subjects and HC [151]. Prostaglandins and other inflammatory molecules influence the activation of trigeminovascular afferents [13]. Among the members of the prostaglandin family, the most studied in migraine mechanisms is prostaglandin-E2 (PGE2). Serum levels of PGE2 are lower in EM patients than in HC [155], but no difference has been reported between EM and HC in PGE2 saliva levels [156]. During a migraine attack, plasma and saliva PGE2 levels increase compared with a pain-free period [48,152,157]. Notably, during migraine attacks, the serum levels of cyclooxygenase 2 (COX-2), an enzyme implicated in the production of PGE2 [158], were higher in migraine patients than in HC [159]. PGE2 serum levels positively correlated with headache frequency in migraine patients [155].

#### 3.3.4. Pentraxin-3 (PTX-3)

PTX-3 is a protein involved in inflammation, innate immunity, and endothelial dysfunction [160,161].

PTX-3 serum levels increase during migraine attacks and are higher in the interictal phase in EM patients than in HC [41,162,163]. Since longer attacks are associated with lower serum levels of PTX-3, PTX-3 is not an indicator of pain intensity [41], while it may be correlated with the length of the disease [162], to suggest that inflammatory processes may change during migraine progression [162]. Increased interictal serum levels of PTX3 were also found in CM compared with HC [38,68,164]. Additionally, CM patients responding to onabotulinumtoxinA showed higher serum levels of PTX-3 than non-responders, which suggests that PTX-3 has a role as a biomarker for treatment selection in CM [68].

#### 3.3.5. IgG

There is some evidence in the literature that dietary intolerances and sensitivities based on IgG cause migraine [165,166,167]. Increased levels of cytokines and IgG antibodies are associated with inflammatory response, which is involved in migraine [168]. The blood IgG levels of EM patients were found higher than those of HC, without any differences between the ictal and interictal phases [166,169]. Xu and colleagues [169] used IgG N-glycopeptide expression to build a migraine prediction model, an intriguing approach that awaits testing in a large population.

#### 3.3.6. Matrix Metalloproteinase-9 (MMP-9)

Matrix metalloproteinases, especially the MMP-9, have drawn attention in relation to migraine discomfort because the increased activity may impact the permeability of the blood–brain barrier [170]. A change in the permeability of the blood–brain barrier, however brief, may occur during a migraine attack [171,172]. Following this hypothesis, the plasma levels of MMP-9 are reported to be significantly higher in EM than in HC in the interictal period and even higher during the headache phase [173,174,175]. These findings, however, were not confirmed by other researchers, who found no changes in MMP-9 levels between subjects with migraine and HC [176,177], suggesting that, although increased MMP-9 levels are intriguing, a more thorough examination is necessary.

It should be noted that while a putative correlation between blood–brain barrier disruption and CSD (related to the aura phenomenon) has been suggested [172,178], multiple studies failed to detect any difference in MMP-9 levels between subjects with aura and those without aura [173,174,175].

### 3.4. Endocannabinoids and Related Lipids

Inflammatory and pain-related mediators are produced by lipids, which are also major energy storage sources [179]. According to Castor et al. [180], aberrant lipid metabolism in CM is linked to alterations in plasma and CSF lipids, which may point to an altered energy equilibrium. Endocannabinoids, endogenous retrograde neurotransmitters with lipid bases widely distributed in peripheral organs and the nervous system, make up the complex cell signaling known as the endocannabinoid system (ECS). The ECS consists of cannabinoid receptors type-1 (CB1) and type-2 (CB2), their endogenous ligands anandamide (AEA) and 2-arachidonoylglycerol (2-AG), and the enzymes involved in their synthesis and degradation. This system is functionally connected with other signaling pathways that include fatty acids, esters, and amides, such as palmitoylethanolamide (PEA). These are congeners of endocannabinoids, which may be synthesized and hydrolyzed by endocannabinoid metabolic enzymes but do not bind to CB receptors [181].

The ECS is implicated in multiple physiological processes and functions, including pain processing and modulation. An increasing amount of evidence suggests a dysregulation of the ECS in migraine [182,183,184]. Concerning the circulating endocannabinoids and related lipids, reduced levels of AEA, 2-AG, and PEA have been reported by some authors in the CSF and platelets of patients with CM and MOH compared with HC [185,186]. Other studies failed to detect significant differences in the plasma levels of AEA and related lipids between EM patients and HC [187,188], suggesting that the deregulation of ECS may be specific to the CM subtype. However, increased levels of PEA were reported in EM patients compared with HC during an experimentally induced migraine attack [188], a finding that was interpreted as a compensatory mechanism. Recently, lower levels of PEA were found in the saliva of migraine subjects compared with control subjects [189], although the study did not report the clinical characteristics of the patients.

A few studies have evaluated the metabolism of endocannabinoids. Cupini et al. [190] found significantly higher activities of fatty acid amide hydrolase (FAAH, the main AEA catabolic enzyme) and the AEA transporter in the platelets of women with EM compared with HC. FAAH and AEA transporter activities were lower in patients with CM and MOH than in EM and HC subjects [191]. Transcriptional changes in ECS components were also reported in the peripheral blood mononuclear cells (PBMCs) of migraine patients compared with controls [192]. These changes were detected peripherally, which makes them amenable for wider adoption to further investigate their role and applicability in the clinical field.

In the NTG-based animal model of migraine, we found increased activity of the endocannabinoid degrading enzymes (FAAH and MAGL) and an increased number of CB receptor binding sites in brain areas [193]. In the same migraine model, AEA administration reduced NTG-induced hyperalgesia during the plantar formalin test and neuronal activation in the trigeminal nucleus caudalis [194].

Notably, treatment with methanandamide, an anandamide synthetic analog, attenuated NTG-induced CGRP increases in plasma, trigeminal ganglia, and the brainstem, and it inhibited dural mast cell degranulation [195]. AEA significantly reduced the neurogenic inflammation caused by dural electrical stimulation in rats [196,197].

Collectively, these results suggest that the ECS is dysfunctional in migraine patients, particularly in those with CM, and call for focused research to validate whether peripheral endocannabinoids and associated lipid levels can be adopted as disease biomarkers.

### 3.5. MicroRNAs

MicroRNAs have recently generated interest as putative biomarkers for migraine [198,199,200]. They are non-coding RNA filaments of 22 or fewer nucleotides involved in modulating physiological circumstances and are associated with several diseases. MicroRNAs interact with the 3’ untranslated region of target mRNAs to promote their degradation and translational repression, which are the two post-transcriptional mechanisms through which they control gene expression [201,202].

There is mounting evidence suggesting that microRNAs are dysregulated in pain conditions, including migraine [66,203,204,205,206]. For instance, Andersen and colleagues found higher MiR-34a-5p levels in serum during migraine attacks, while miR-382-5p levels were higher in the interictal period [204]. Furthermore, juvenile migraine patients receiving treatment had a lower peripheral expression of miR-34a-5p in their saliva, suggesting a potential role in therapeutic response prediction [198]. MiR-30a expression was lower in migraine patients with and without aura serum [205], whereas miR-155 expression was higher during interictal phases [206].

Recently, a link between some microRNAs and phenotype and migraine severity has been proposed. Specifically, the interictal expression of miR-382-5p and miR-34a-5p was significantly higher in the PBMCs of patients with CM and MOH [66]. In contrast, a previous study failed to detect any significant difference in the expression of the same microRNA gene in these cells. The contrasting data are likely due to the small number of patients used [207].

In the preclinical area, there are no data about circulating microRNA levels, but we showed that NTG induced an increase in the miR-155-5p, miR-34a-5p, and miR-382-5p expression in specific areas of the central nervous system of rats. This increase was significantly attenuated by the CGRP antagonist olcegepant [56].

More clinical and mechanistic investigations are required to clarify the physiological roles played by microRNA in migraine and to validate them as migraine biomarkers [208].

### 3.6. Mediators of Endothelial/Vascular Functions

Some studies suggest an alteration in the endothelial function in migraine [34,209,210,211] and a relationship between migraine and vascular risk [212].

The mediators associated with endothelial functions whose blood levels appear to change in migraine are discussed below.

#### 3.6.1. Endothelial Progenitor Cells (EPCs)

EPCs are circulating cells, considered markers of endothelial function [213], involved in vascular homeostasis and integrity [214].

EM patients showed reduced numbers and functions of EPCs from peripheral blood than HC [34,215,216], without any difference between EM patients with and without aura [34,216]. Lower EPC counts were also found during headache attacks, compared with the pain-free state, and this further decreased with the longer duration of the disease [34]. These findings suggest that migraine patients experience a long-term change in endothelial function (with a decreased ability to repair the endothelium) [34]. Additionally, the reduction in and dysfunction of EPCs in migraine raise the possibility that migraine and cardiovascular risk are related [215,216]. Data in animal models support the suggestion that triptans and β-blockers interact with the endothelial cell component of the blood vessel to produce anti-hyperalgesia [217].

#### 3.6.2. Endothelin-1 (ET-1)

ET-1 is a potent vasoconstrictor produced by vascular endothelial cells [218]. ET-1 may be involved in the onset of the aura phenomenon and the ensuing migraine headache because of the vascular connection in the CSD-related pathways [219]. Higher ET-1 plasma levels have been reported during the ictal phase in EM patients than in HC, particularly in the early stages of the attack [220,221,222,223,224].

By contrast, as regards the interictal phase, the findings are conflicting [221,222,225,226,227]. Conflicting results with either an increase or a decrease in ET-1 plasma levels were reported after NTG administration in rats [108,228]. Moreover, ET-1 was reported to induce CSD in rats [229,230]. However, intravenous ET-1 in migraine with aura patients failed to provoke migraine aura symptoms and did not induce any headache [231]. Furthermore, ET-1 plasma levels did not differentiate between patients with and without aura [221,224,226].

Thus, we suggest that ET-1 function is more likely related to the vascular tone alterations that are noticed when migraine attacks first start, most likely those that occur at early time points [219]. However, based on the available evidence, ET-1 does not represent a reliable indicator for separating patients with aura from those without aura.

#### 3.6.3. Homocysteine

Homocysteine (Hcy) is a simple sulfur-containing molecule that differs from the amino acid cysteine in the presence of a single adjunctive methylene group. Hcy is synthesized in humans from methionine, its precursor, throughout a complex metabolic pathway that involves several essential enzymes and co-factors [232]. Different degrees of hyperhomocysteinemia are usually present alongside the absence of one or more enzymes and co-factors [233,234].

Hcy serum levels were higher in migraine patients, in particular in those with aura, compared with HC, although other studies report lower serum Hcy levels in patients without aura [235,236,237,238,239,240]. Furthermore, Hcy plasma levels were lower in females than in male patients with migraine [241]. Several other studies failed to detect a significant difference between migraine patients and HC [242,243,244,245]. Concerning CSF, according to one study [246], patients with aura have higher CSF Hcy levels than patients without aura. The elevated levels of Hcy enhance migraine symptoms such as increased cortical excitability, mechanical allodynia, photophobia, and anxiety in rats [234,247].

Higher-than-normal Hcy levels have been linked to an increased risk of vascular events, whereas lower-than-normal levels seem to be protective. Further research is needed to disentangle a possible link between Hcy levels in migraine and vascular risk.

### 3.7. Other Biomarkers

These represent potential biomarkers for migraine with limited/conflicting data from human studies and a lack of preclinical evidence.

#### 3.7.1. Tryptophan and Kynurenine

Tryptophan is an amino acid essential for the biosynthesis of different proteins, including 5-HT and melatonin. About 95% of tryptophan is metabolized by the kynurenine pathway, which generates neuroactive compounds that interact with glutamate receptors [248].

Clinical and preclinical evidence suggests a depressed kynurenine pathway in migraine [110,249,250,251]. Indeed, tryptophan, and most of the kynurenine pathway metabolites, were found to be decreased interictally in the plasma of EM patients compared with HC subjects [110] and ictal period [252]. Others, however, found no difference in plasma tryptophan levels in EM [253] or even an increase in EM and CM patients’ plasma/serum levels when compared with HC [254,255]. Notably, kynurenine metabolites in serum decrease in CM patients [254]. These findings and the neurobiological link between tryptophan, kynurenine, and glutamate call for more studies on tryptophan metabolites to test whether they play any role as biomarkers.

#### 3.7.2. Melatonin

The fundamental relevance of melatonin in the regulation of circadian rhythms and sleep is widely known. Melatonin may also have a crucial involvement in headache disorders [256]. Several studies have reported lower plasma/serum and urine melatonin levels in migraine patients (EM and CM) than in HC, especially during a migraine attack. Melatonin could also be effective in treating migraines by lowering the number of days with headaches per month [257,258,259,260].

#### 3.7.3. Growth Factors

Growth factors are a broad class of secreted proteins known to exert a key role in regulating cell survival, growth, and differentiation; some of them, such as nerve growth factor (NGF), brain-derived neurotrophic factor (BDNF), and neurotrophin 4/5, can enhance pain in some circumstances [261,262,263]. The most studied within the migraine field are the neurotrophins NGF and BDNF; however, it should be noted that changes in other growth factors were also reported in migraine sufferers. For instance, CM and EM patients showed higher neurotrophin 4/5 plasma levels than HC [264]. Notably, the authors did not detect any difference in glial-cell-line-derived neurotrophic factor (GDNF), while Sarchielli et al. [265] reported lower levels of GDNF in the CSF of CM patients tested during the interictal period compared with HC. Regarding NGF, the literature provides conflicting findings on the amounts of circulating levels. Patients with EM tested interictally had lower NGF plasma levels than HC [155,266]. In comparison, Martins et al. [264] showed no difference in NGF plasma levels between migraine patients (EM and CM) and HC. Other studies reported higher CSF/plasma/saliva levels of NGF in CM patients compared with HC [29,33,267]. Interestingly, NGF levels seem to positively correlate with headache frequency [29,155]. Regarding BDNF, higher levels were reported in the CSF of CM patients compared with HC [267]. Additionally, BDNF plasma levels were higher in EM subjects during the ictal period compared with the interictal one and HC [268,269]. Notably, other studies found lower plasma/serum levels of BDNF in EM and CM compared with HC [155,266] and lower levels in CM compared with EM [264].

In conclusion, the heterogeneity of findings precludes at this moment the possibility to suggest a role for growth factors as migraine biomarkers. Migraine pathogenesis is associated with oxidative stress by altering cerebral blood flow [270,271].

#### 3.7.4. Mediators of Oxidative Stress

Oxidative stress can be induced by common migraine triggers. These mechanisms include, for instance, a high rate of energy production by the mitochondria, toxicity, calcium excess, excitotoxicity, and neuroinflammation, depending on the stimulus [272]. Increased levels of reactive oxygen and nitrogen species increase vulnerability to oxidative compounds and the reduction in antioxidative defense. Selenoproteins such as glutathione peroxidases (GPx), thioredoxin reductases (TrxRs), or selenoprotein P (SelP) with antioxidant activity are indeed crucial for maintaining the physiology of neurons and glial cells [273]. Malondialdehyde (MDA) is a product of lipid peroxidation and has been widely used as a biomarker of oxidative stress [274,275]. Nitric oxide (NO) is a well-known oxidant/vasodilator with a critical function in migraine pathophysiology, and it is considered an indicator of nitrosative stress [276]. Patients with migraine show higher serum levels of MDA [277]; this finding confirmed the similar results obtained previously by Togha et al. [278], who reported higher interictal serum levels of MDA in both CM and EM subjects compared with HC. In the same study, the authors reported higher NO serum levels in both groups of migraine patients compared with HC [278]. Interestingly, higher levels of both MDA and NO metabolites were detected in the platelets of migraine patients evaluated ictally [278]. The higher levels of NO metabolites and nitrites were also found in plasma, supporting NO modulatory role in biological processes, particularly vasodilation [48]. Catalase and superoxide dismutase are antioxidant enzymes less studied in the migraine area. Their serum levels were lower in CM subjects when compared with EM or HC subjects in several studies [278,279].

#### 3.7.5. Apolipoprotein E

(ApoE) ApoE is implicated in lipid transport and metabolism, and it is involved in different neurological and vascular disorders [280,281], including migraine. Indeed, serum ApoE protein levels were higher in EM patients than in HC [45], particularly during migraine attacks [45,282]. Accordingly, ApoE polymorphisms were associated with an increased risk of headaches [283] and migraine [284]. Although there have only been a few studies reporting changes in circulating levels, ApoE appears to be a potential biomarker for migraine diagnosis and is worth further investigating [285].

## 4. Fitting Together the Pieces of a Complex Picture

Migraine disease consists of recurrent attacks that display a cyclic profile made of different phases, culminating in a full-blown attack, characterized by pain and multiple associated symptoms [5]. Multiple systems and mediators are involved in different phases of the attack and thus in the diverse clinical manifestation of the disease itself. Several studies have investigated a multitude of circulating signaling molecules/biomarkers linked to migraine pathogenesis. Table 1 outlines an analytical list of the alterations documented in preclinical and clinical studies. The possible interactions between pathways and these signaling molecules are summarized in Figure 1.

TVS activation is generally accepted as one of the primary pathophysiological processes in migraine pathophysiology [8], although it is still uncertain whether TVS activation is of central or peripheral origin. The identification of the biological agents involved in migraine pathogenesis is a further pertinent question. As demonstrated by a number of clinical and preclinical investigations, the function of CGRP is undeniable in this regard. However, due to the complexity of migraine disease, it is more likely that multiple factors act in concert together with the neuropeptide CGRP. CGRP is released by the peripheral and central terminals of trigeminal ganglion neurons and has the ability to cause nociception, vasodilation, and neurogenic inflammation [286]. Considering its significant impact on migraine pathophysiology, researchers have worked hard to create medications that either target CGRP or its receptor complex. The development and use of monoclonal antibodies directed against the neuropeptide CGRP or its receptor are the most recent achievement in the field of migraine [287]. However, despite the success displayed by such monoclonal antibodies, there is still a great percentage of migraine patients for whom the beneficial effects are minimal or completely absent, which underscores the involvement of other biological players in migraine generation and maintenance besides CGRP [288,289,290]. An important role is played by NO. CGRP and other peptides, once released from activated trigeminal fibers in the trigeminovascular space, act on the vascular smooth muscle cells inducing vasorelaxation [25] and endothelial cells to promote NO release [291]. This sequence of events is supported by both human and animal studies [27,48,291,292,293,294,295,296,297,298,299,300,301,302]. Protein ApoE may also be required for NO production in addition to the pathways indicated above [303]. Specifically, ApoE polymorphisms can alter arginine uptake, resulting in increased NO, via a NOS-independent mechanism [304], but it also may influence the expression of cytokines [305] and CGRP [27]. Besides the NO-induced pathways, CGRP release may promote the production of inflammatory mediators such as cytokines [306]. In agreement, data from the NTG animal model of migraine report higher protein levels or gene expression of proinflammatory cytokines (e.g., IL-1β, IL-6, and TNF-α) in the peripheral blood and trigeminovascular areas [56,302,307,308]. These alterations have been confirmed in the peripheral blood of migraine patients [130,131]. Elevated serum levels of TNF-α have also been linked with endothelial dysfunction, which is reportedly present in migraine patients [211,309,310]. The increased plasma levels of ET-1 in migraine patients [219] can cause a release of CGRP [311], NO [312], and proinflammatory mediators [313]. Other neuropeptides, including SP, VIP, and PACAP may contribute to trigeminovascular activation [23,314]; meanwhile, increased inflammation may cause NGF and BDNF release, which in turn affect nociceptive pathways [264]. NGF may induce hyperalgesia by enhancing the production and release of SP and CGRP, via the activation of the transient receptor potential vanilloid 1 (TRPV1) channel [264,315,316]. Some microRNAs associated with inflammation, such as miR-382-5p and miR-34a-5p, were altered in CM patients. These microRNAs target genes involved in anti-nociceptive and anti-inflammatory mediators regulation [200]. The intriguing concept related to these microRNAs is that not only can they modulate the molecular targets relevant to migraine pain, but they can also be stimulated by molecules whose levels are increased in migraine [317,318,319,320]. An alteration in the glutamatergic neurotransmission leads to excitotoxicity and neuronal hyperexcitability [321]. On the other hand, a downregulation of the kynurenine pathway may increase CGRP and PACAP levels, thus enhancing the generation of migraine attacks [322]. Among the numerous systems and pathways outlined above, all of which are connected and reciprocally influenced, the ECS is deeply entangled with most of them [323,324,325]. Indeed, ECS and related lipids are involved in migraine-related mechanisms, such as the inflammatory pathways, neuropeptidergic and neurotransmitter signaling. AEA can inhibit NO and CGRP-induced dural vasodilatation [196] and can desensitize the TRPV1 channels [326], whose activation on the trigeminal fibers promotes the release of CGRP [197,327]. Moreover, PEA may indirectly lead to the downregulation of proinflammatory transcription factors such as AP-1 and NF-κB [328]. ECS may cooperate differently with other migraine-related pathways, such as the kynurenine pathway [323,324], thus causing the modulation of glutamatergic signaling.

## 5. Considerations and Perspectives

The search for a potential biomarker for migraine has persisted for a long time, and the potential molecules implicated in disease pathogenesis and chronification have been investigated in saliva, blood, and CSF. However, even in the hopeful case of CGRP and other biomarkers, their clinical use may be affected by a variety of biological factors, such as phases of the menstrual cycle, and non-biological factors, including the lack of standardized protocols and methodologies. Thus, the non-specificity and non-sensitivity of biomarker change remain a problem in migraine. At present, studies suggest that the action of the CGRP is pivotal in the trigeminovascular complex, but other neurobiological factors associated with it may offer alternative possibilities in precision medicine. The exact mechanisms through which CGRP initiates and sustains the other pain and inflammatory mediators in a reciprocal loop are yet to be defined, although recently, it was proposed that SFK activity plays a pivotal role in facilitating the crosstalk between CGRP and cytokines by transmitting CGRP receptor/protein kinase A signaling in TG and trigeminovascular sensitization [329]. An evaluation of multiple panels of biomarkers, including CGRP and other neuropeptides, microRNAs, and proinflammatory peptides could be useful to identify the signatures of migraine patients and to develop personalized therapy.

The wealth of clinical and preclinical data on the role of sexual hormones in migraine [330], as well as the output of in vitro and in vivo studies evaluating the mediation of prolactin [73], calls for future fundamental and clinical studies aimed at investigating certain aspects of sex-related responses and differences between females and males in laboratory settings and in humans.

## Figures and Tables

**Figure 1 ijms-24-05334-f001:**
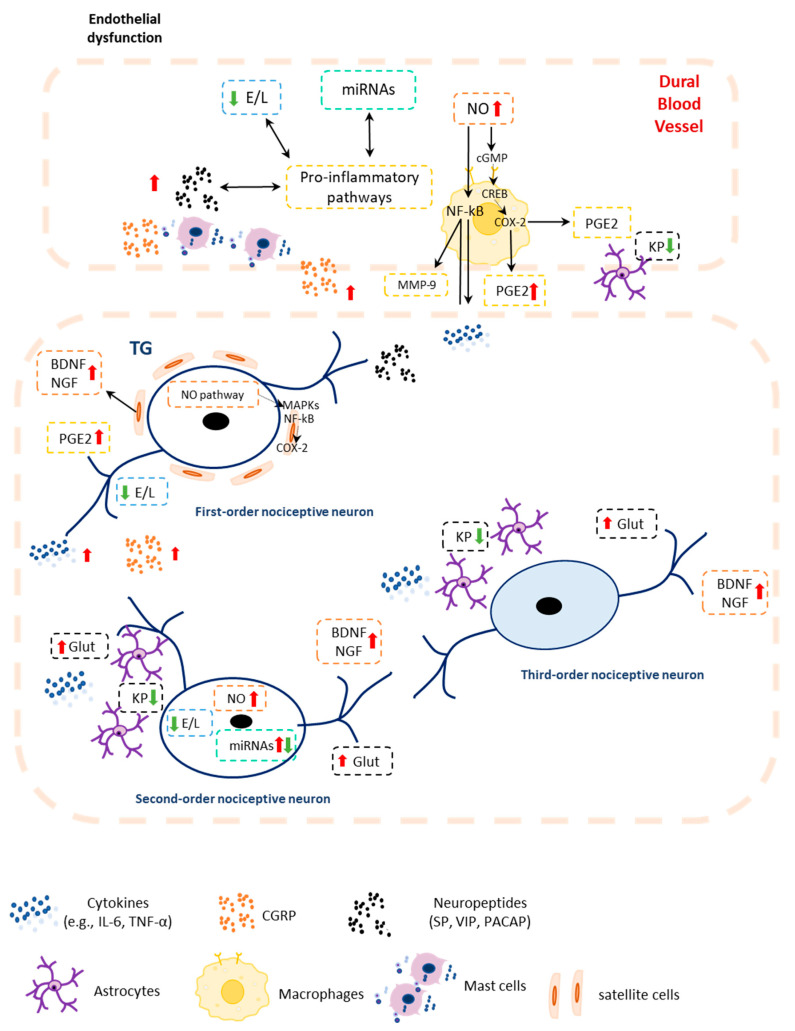
Mechanisms underlying migraine pathophysiology: Reciprocal interactions among the systems and pathways whose components are altered in migraine patients and potential molecular interactions within the neuropeptides (i.e., CGRP)–inflammation–ECS in neuronal and non-neuronal cells. Red arrow: increase; green arrow: decrease; BDNF: brain-derived neurotrophic factor; cAMP: cyclic AMP; cGMP: cyclic GMP; CGRP: calcitonin gene-related peptide; COX-2: cyclooxygenase-2; CREB: cyclic AMP response element-binding protein; ECS: endocannabinoid system; E/L: endocannabinoids and related lipids; Glut: glutamatergic signaling; IL-6: interleukin 6; KP: kynurenic pathway; MAPKs: mitogen-activated protein kinases; MMP-9: matrix metallopeptidase 9; NF-kB: nuclear factor kappa B; NGF: nerve growth factor; NO: nitric oxide; PACAP: pituitary adenylate-cyclase-activating polypeptide; PGE2: prostaglandin E2; SP: substance P; TNF-α: tumor necrosis factor-alpha.

**Table 1 ijms-24-05334-t001:** Biomarker evaluations and main findings.

Biomarkers	Clinical Sample	Preclinical Sample	Clinical Findings	Preclinical Findings	Key Conclusions
Neuropeptides					
CGRP	Blood (plasma, serum), Saliva, CSF	Blood (plasma, serum)	↑ Levels in EM and CM patients during interictal and ictal phases [28,29,30,31,32,33,34,35,36,37,38,39,40,41,42,47,48,49,54]; Index of good response to onabotulinumtoxinA treatment and detoxification in CM or MO-CM patients [36,52,65,66,68]; Unchanged levels in EM and CM patients during interictal and ictal phases [42,43,44,45,46,52,63]	↑ in rodents and cat [55,56,57,58,59,60,61,62]	Potential diagnostic biomarker and potential biomarker for prediction of response to treatments
Substance P	Blood (plasma, serum)	Blood (plasma, serum)	↑ Levels in EM patients during interictal and ictal phases and interictal CM [32,33,79] Unchanged levels in EM patients and HC [80,81]	↑ levels in rat [62]	
PACAP	Blood (plasma, serum)	Blood (plasma)	↑ Levels in EM patients during interictal and ictal phases [42,86,87,89] or ↓levels in EM patients during interictal phase [87,88]↑ levels in CM patients during interictal phase [42] Unchanged in EM or CM during interictal or vs. HC [42,85]	↑ levels in rat and cat [57,61,62,90]	Potential Therapeutic Biomarker
VIP	Blood (plasma)	Blood (plasma)	↑Levels EM and CM patients during interictal and ictal phases [31,36,42,65,81,85,86] Correlation with autonomic symptoms [81,94] and cranial parasympathetic symptoms [95] Index of good response to triptan treatment in EM [94]	↑ levels in rat [61,62]	
NPY	Blood (plasma) CSF	Blood (plasma)	↑ Levels in CSF [100] or ↓ levels in plasma in CM and EM patients during ictal phase migraine patients [102] Unchanged levels in EM patients during ictal and interictal phases and vs. HC [81,101]	↑ levels in rat [61]	
Classic Neurotransmitters					
Glutamate	Blood (plasma) Saliva CSF	Blood (serum)	↑ Plasma and salivary levels in EM and CM patients vs. HC during interictal and ictal phase [106,114,115,116,117,118,119] ↑ CSF levels in EM and CM vs. HC during ictal phase [30,120,121]	↑ levels in rat [122]	
Inflammatory Mediators					
Cytokines	Blood (plasma, serum), CSF	Blood (serum)	↑ IL-6, TNF-α, IL-1β, TGF-β1 levels in EM patients during interictal phase [132,138,139,140] ↑ IL-6, TNF-α levels in CM patients during ictal and interictal phase [134,144,145]	↑ IL-6, TNF-α, IL-1β levels in rat [56]	
Other Potential Biomarkers					
Endocannabinoids and related lipids	Blood (platelets, plasma), Saliva, CSF	Data available only for brain areas	↑ PEA levels in EM patients during ictal phase induced by nitroglycerin [188] ↓ Salivary PEA levels in migraine patients [189] ↓ AEA, 2-AG, and PEA levels in CM and MOH patients [185,186] AEA and related lipid levels unchanged during the interictal phase in EM patients vs. HC [187,188] ↑FAAH and AEA transporter activities in woman EM patients [190] ↓FAAH and the AEA transporter activity in CM and MOH patients [191]	Increased levels of the degrading enzymes FAAH and MAGL [193]	
MicroRNAs	Blood (plasma, serum)	Data available only for brain areas	↑ miR-34a-5p levels in CM during ictal phase and in CM and MOH patients during interictal phase [66,204] ↑miR-382-5p levels CM during ictal phase and CM and MOH patients during interictal phase [66,204] ↓ miR-30 in EM patients [205] ↑ miR-155 in EM patients in interictal phase [206]	↑ miR-155-5p, miR-34a-5p, and miR-382-5p in several brain areas. Inhibitory effect of a CGRP antagonist [56]	
ET-1	Blood (plasma)	Blood (plasma)	↑ levels in EM patients in the early stages of the attack [220,221,222,223,224] Conflicting results in interictal phase [221,222,225,226,227]	↑ or ↓ levels in rodents [108,228]	
Tryptophan and Kynurenine Metabolism	Blood (plasma, serum)	CSF	↓ Levels in EM patients during interictal and ictal phases [110,252] ↓ levels only of kynurenine metabolites in CM patients during interictal phase [254] ↑ levels in EM and CM patients vs. HC during interictal phase [254,255] Unchanged levels in EM patients during interictal phase [253]	↓ levels in rat [251]	

## Data Availability

Not applicable.

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
