# Peer review of "Biomarkers of Migraine: An Integrated Evaluation of Preclinical and Clinical Findings"

_ijms, 2023, doi:10.3390/ijms24065334_

Round 1
Reviewer 1 Report
This paper nicely reviewed current progress on potential biomarkers in migraine pathogenesis by comparing available clinical and preclinical data from circulating biomarkers perspective. The biomarkers range from the well-known drug target, CGRP to cytokines, neurotransmitters and microRNAs. I have following minor points for improvement.
1. Multiple neurotransmitters are known to play crucial roles in migraine mechanisms. In section 3.5, glutamate was the only neurotransmitter that was reviewed, which then seems to be incomprehensive. Some peptides, e.g., CGRP, one type of neurotransmitters, is reviewed in a different section (3.1), which seems a little confusing.
2. A few research papers recently show that PKA and Src family kinases contribute to the cross talk between CGRP and cytokines in trigeminal ganglion, which could partially explain exact mechanism by which CGRP initiates and sustains other pain and inflammatory mediates as stated in Section 5. Perhaps to briefly introduce here.
3. Figure 1B could be improved indicating different types of potential markers in different color. Are these biomarkers in same cell type?
Author Response
Comments and Suggestions for Authors
This paper nicely reviewed current progress on potential biomarkers in migraine pathogenesis by comparing available clinical and preclinical data from circulating biomarkers perspective. The biomarkers range from the well-known drug target, CGRP to cytokines, neurotransmitters and microRNAs. I have following minor points for improvement.
We are grateful to the Referee’s for the positive comment.
Multiple neurotransmitters are known to play crucial roles in migraine mechanisms. In section 3.5, glutamate was the only neurotransmitter that was reviewed, which then seems to be incomprehensive. Some peptides, e.g., CGRP, one type of neurotransmitters, is reviewed in a different section (3.1), which seems a little confusing.
R
In order to address the issue, in the revised version we modified the title of the Section where we mention briefly the classic neurotransmitters and glutamate. It is now called “Classic Neurotransmitters”, to avoid confusion with the “Neuropeptides” section.
A few research papers recently show that PKA and Src family kinases contribute to the cross talk between CGRP and cytokines in trigeminal ganglion, which could partially explain exact mechanism by which CGRP initiates and sustains other pain and inflammatory mediates as stated in Section 5. Perhaps to briefly introduce here.
R
As suggested by the Referee, we added additional information about PKA and Src family kinases in Section 5.
Figure 1B could be improved indicating different types of potential markers in different color. Are these biomarkers in same cell type?
R
We are grateful to the Referee for this suggestion, we improved Figure 1 accordingly.

Reviewer 2 Report
Migraine is a common neurological disorder characterized by recurrent headaches. According to the Global Burden of Diseases, migraine represents the third most frequent and the second most disabling condition in humans, affecting about 16% of the world's population.
One of the main obstacles to the accurate diagnosis and treatment of migraine is the lack of reliable biomarkers. The search for biomarkers of migraine has been going on for a long time. Specific and individualized biomarkers of migraine can significantly facilitate the proper diagnosis of migraine and help to explore the pathophysiology and new treatment strategies.
The potential molecules involved in the pathogenesis and chronicity of the disease have been studied mostly in saliva, blood, and cerebrospinal fluid.
The purpose of this review is to summarize the potential migraine biomarkers and to discuss their role in the pathogenesis of the disease. Moreover, the authors discuss the potential issues affecting biomarker analysis, such as how to deal with bias and confounding data.
The data originating from both clinical and pre-clinical studies on migraine were retrieved from the PubMed/MEDLINE database, covering the period from 1988 to 2022.
The data in the manuscript come from studies conducted on adult patients with EM (with or without aura) or CM (including the MOH subgroup). In the case of preclinical studies, articles dealing with animal models (rats or mice) reproducing one or more pathophysiological features associated with migraine pain were considered. Of these clinical and preclinical studies, only those that reported changes in circulating (ie, blood, cerebrospinal fluid, and urine) biomarkers were considered.
The review discusses several potential biomarkers and their possible role in migraine. The authors also point out that even with the promise of CGRP and other biomarkers, many biological and non-biological factors may affect their clinical application, including the lack of standardized protocols and methods.
The topic is timely and may attract much attention. The study is well-conducted and designed.
Minor problems:
1. Several space and typing errors are present in the manuscript. Please, review the text carefully and correct these errors.
E.g.:
Line 46 [2] . In
Line 77 [315] , whose
Line 78 [10,12] .
Line 109 [23].The
Line 128 [32,38,46-48] ;.
Line 130 [43,49].Similarly
Line 154 [55-60].By
Line 148 [53,54,58,62] . In
Line 191 [31] . Nicolodi
Line 195 [75] , a
Line 251 system[90]
Line 315 [117-119] . Interestingly
Line 381 source[147]. ; al. , aberrant
Line 559 [237] (missing full stop)
Line 600 [257,258] . Of
Line 703 [315] , whose
2. The complete form of the abbreviation appears in several places, e.g.: lines 40-41 and lines 76-77 episodic and chronic migraine.
3. Although they are known abbreviations, it would still be worthwhile to use the full form in the first place of appearance and then use the abbreviated form. E.g.: VIP, NPY, SP (lines 102-103).
4. Reference formatting: Line 251 [90] [89] - The formatting of the references differs here compared to the others. Elsewhere the references are enclosed in parentheses.
5. Before 3.6.2. and 5. chapter there is a double blank line.
Author Response
REPORT 2
Comments and Suggestions for Authors
Migraine is a common neurological disorder characterized by recurrent headaches. According to the Global Burden of Diseases, migraine represents the third most frequent and the second most disabling condition in humans, affecting about 16% of the world's population.One of the main obstacles to the accurate diagnosis and treatment of migraine is the lack of reliable biomarkers. The search for biomarkers of migraine has been going on for a long time. Specific and individualized biomarkers of migraine can significantly facilitate the proper diagnosis of migraine and help to explore the pathophysiology and new treatment strategies. The potential molecules involved in the pathogenesis and chronicity of the disease have been studied mostly in saliva, blood, and cerebrospinal fluid. The purpose of this review is to summarize the potential migraine biomarkers and to discuss their role in the pathogenesis of the disease. Moreover, the authors discuss the potential issues affecting biomarker analysis, such as how to deal with bias and confounding data. The data originating from both clinical and pre-clinical studies on migraine were retrieved from the PubMed/MEDLINE database, covering the period from 1988 to 2022. The data in the manuscript come from studies conducted on adult patients with EM (with or without aura) or CM (including the MOH subgroup). In the case of preclinical studies, articles dealing with animal models (rats or mice) reproducing one or more pathophysiological features associated with migraine pain were considered. Of these clinical and preclinical studies, only those that reported changes in circulating (ie, blood, cerebrospinal fluid, and urine) biomarkers were considered. The review discusses several potential biomarkers and their possible role in migraine. The authors also point out that even with the promise of CGRP and other biomarkers, many biological and non-biological factors may affect their clinical application, including the lack of standardized protocols and methods. The topic is timely and may attract much attention. The study is well-conducted and designed.
Minor problems:
Several space and typing errors are present in the manuscript. Please, review the text carefully and correct these errors.
E.g.: Line 46 [2] . In; Line 77 [315] , whose; Line 78 [10,12] .; Line 109 [23].The, Line 128 [32,38,46-48] ;.
Line 130 [43,49].Similarly, Line 154 [55-60].By Line 148 [53,54,58,62] . In, Line 191 [31] . Nicolodi, Line 195 [75] , a Line 251 system[90], Line 315 [117-119] . Interestingly Line 381 source[147]. ; al. , aberrant
Line 559 [237] (missing full stop) Line 600 [257,258] . Of Line 703 [315] , whose
R
We are grateful for the words of appreciation of the Referee. We have corrected the list of errors.
The complete form of the abbreviation appears in several places, e.g.: lines 40-41 and lines 76-77 episodic and chronic migraine.
R
We thank the Referee spotting these inconsistencies that have been corrected in the revised version.
Although they are known abbreviations, it would still be worthwhile to use the full form in the first place of appearance and then use the abbreviated form. E.g.: VIP, NPY, SP (lines 102-103).
R
We have added the full form of all abbreviations when first used, as suggested by the Referee.
Reference formatting: Line 251 [90] [89] - The formatting of the references differs here compared to the others. Elsewhere the references are enclosed in parentheses.
R
The Reference formatting has been revised for consistency.
Before 3.6.2. and 5. chapter there is a double blank line.
R
The double blank line has been deleted.

Reviewer 3 Report
Summary: This manuscript is a narrative review summarising the pre-clinical and clinical work surrounding the elevation or reduction of molecules, such as neuropeptides, inflammatory markers and vascular markers, in migraine patients and how these compare to healthy controls. This review summarises the data in the table and highlights which markers may be useful for migraine diagnosis, which may not and which need further work.
Major points:
Experimental design and results:
1) It is worth expanding the neuropeptide section to discuss all the members of the calcitonin family of peptides, not just CGRP and amylin. Other members of the calcitonin family of peptides have been reported to be elevated in migraineurs, such as pro-calcitonin. Several studies report that adrenomedullin can induce migraine-like attacks and calcitonin is reported to have some analgesic abilities in migraine. Additionally, the authors should include the human clinical studies demonstrating that the amylin analogue, pramlintide, and adrenomedullin can induce migraine-like attacks to further highlight their potential role. Overall, it should be mentioned that although there is limited research on these peptides, they could be potential biomarkers and further research is needed.
2) The authors frequently discuss “animal models for migraine”. Please describe each model when mentioned in the text, e.g., lines 295-296, which state “rats treated with NTG”. This is because there are many different migraine models or models of migraine-like symptoms, such as photophobia, grimace and pain, and this context is important for interpreting the studies.
3) Women are disproportionately affected by migraine. There is a complete absence of discussion surrounding sex hormones and migraine in this review. Several studies have linked changes in CGRP to the menstrual cycle, menopause etc. Additionally, several recent meta-analysis reviews have highlighted the potential role of prolactin in migraine and changes in prolactin levels. There are likely other possible markers that have been linked to sex hormones and migraine that could be discussed. This should be addressed with an additional section regarding sex hormones.
Minor points:
1) There are many formatting errors throughout the document, such as additional spaces or lack of spaces. Additionally, some paragraphs have formatting where each sentence is on a new line and should be combined into a traditional paragraph.
2) Please correct “As regards” at the start of paragraphs to “regarding”.
3) The authors talk about the variability of PACAP38 results. Could the authors look into and comment on what they think the source of this variability might be? Is it due to the timepoint assayed? The type of kit was used? The number of n in the study? Etc
4) Page 5, line 27, what is common or classic migraine? Compared to what?
5) Page 6, lines 251-253, the authors discuss the link between VIP and CGRP expression and middle meningeal artery size. Please explain why this is relevant, as this is not directly related to VIP as a biomarker but is more mechanistic.
6) Section 3.2.1, Is there any data for C-reactive protein in migraine? If so, please add it to the inflammation and immunity section.
7) Page 7, line 313, “the involvement of adipocytokines in migraine is not surprising” – using is a bit too definitive based on the current data as there doesn’t yet appear to be a clear mechanistic link established. Authors should change the wording to “would not be surprising” or something to that effect.
8) Section 3.2.3, Is there any link between prostaglandin, when elevated during menstruation, and migraine?
9) Authors should combine sections 3.5 and 3.5.1
10) Page 11, lines 537-538, the authors use “compared to control” twice in the sentence.
Author Response
REPORT 3
Comments and Suggestions for Authors
Summary: This manuscript is a narrative review summarising the pre-clinical and clinical work surrounding the elevation or reduction of molecules, such as neuropeptides, inflammatory markers and vascular markers, in migraine patients and how these compare to healthy controls. This review summarises the data in the table and highlights which markers may be useful for migraine diagnosis, which may not and which need further work.
Major points:
Experimental design and results:
It is worth expanding the neuropeptide section to discuss all the members of the calcitonin family of peptides, not just CGRP and amylin. Other members of the calcitonin family of peptides have been reported to be elevated in migraineurs, such as pro-calcitonin. Several studies report that adrenomedullin can induce migraine-like attacks and calcitonin is reported to have some analgesic abilities in migraine. Additionally, the authors should include the human clinical studies demonstrating that the amylin analogue, pramlintide, and adrenomedullin can induce migraine-like attacks to further highlight their potential role. Overall, it should be mentioned that although there is limited research on these peptides, they could be potential biomarkers and further research is needed.
R
We are grateful to the Referee for this suggestion. In the revised version, we provided additional information about the calcitonin family of peptides now in the CGRP section (Page 4, line 176).
The authors frequently discuss “animal models for migraine”. Please describe each model when mentioned in the text, e.g., lines 295-296, which state “rats treated with NTG”. This is because there are many different migraine models or models of migraine-like symptoms, such as photophobia, grimace and pain, and this context is important for interpreting the studies.
R
According to Referee’s suggestion, we have provided additional information regarding the migraine models mentioned in the text.
Women are disproportionately affected by migraine. There is a complete absence of discussion surrounding sex hormones and migraine in this review. Several studies have linked changes in CGRP to the menstrual cycle, menopause etc. Additionally, several recent meta-analysis reviews have highlighted the potential role of prolactin in migraine and changes in prolactin levels. There are likely other possible markers that have been linked to sex hormones and migraine that could be discussed. This should be addressed with an additional section regarding sex hormones.
R
Although sexual hormones and prolactin are likely to play a role in migraine pathophysiology, their potential use as migraine biomarkers is still too limited. We have mentioned this gap in the conclusive paragraph.
Minor points:
There are many formatting errors throughout the document, such as additional spaces or lack of spaces. Additionally, some paragraphs have formatting where each sentence is on a new line and should be combined into a traditional paragraph.
R
We have revised text formatting throughout the manuscript.
Please correct “As regards” at the start of paragraphs to “regarding”.
R
The Referee’s suggestion has been accepted and applied to the text.
The authors talk about the variability of PACAP38 results. Could the authors look into and comment on what they think the source of this variability might be? Is it due to the timepoint assayed? The type of kit was used? The number of n in the study? Etc
R
According to the Referee’s request, some more details were inserted in this section (Page 5, line 242).
Page 5, line 27, what is common or classic migraine? Compared to what?
R
We have revised the sentence by applying the updated terminology for migraine subtypes and the mistake has been corrected.
Page 6, lines 251-253, the authors discuss the link between VIP and CGRP expression and middle meningeal artery size. Please explain why this is relevant, as this is not directly related to VIP as a biomarker but is more mechanistic.
R
We have elaborated the sentence in order to meet the Referee’s request.
Section 3.2.1, Is there any data for C-reactive protein in migraine? If so, please add it to the inflammation and immunity section.
R
Basic information on C-reactive protein has been inserted in this section (Page 7, line 333).
Page 7, line 313, “the involvement of adipocytokines in migraine is not surprising” – using is a bit too definitive based on the current data as there doesn’t yet appear to be a clear mechanistic link established. Authors should change the wording to “would not be surprising” or something to that effect.
R
The wording has been modified, as suggested.
Section 3.2.3, Is there any link between prostaglandin, when elevated during menstruation, and migraine?
R
According to the Referee’s request, some more details and suggestions about the link between prostaglandin and migraine were inserted in this section.
Authors should combine sections 3.5 and 3.5.1
R
The sections have been combined.
Page 11, lines 537-538, the authors use “compared to control” twice in the sentence.
R
The mistake has been corrected.
